# How social learning shapes the efficacy of preventative health behaviors in an outbreak

**Simon Carrignon**[1]*, **R. Alexander Bentley**[1], **Matthew Silk**[2], **Nina H. Fefferman**[3,4]

**1** Department of Anthropology and Center for the Dynamics of Social Complexity (DySoC), University of Tennessee, Knoxville, TN, United States of America, **2** Centre for Ecology and Conservation, University of Exeter, Exeter, United Kingdom, **3** Department of Ecology and Evolutionary Biology, University of Tennessee, Knoxville, TN, United States of America, **4** Department of Mathematics, University of Tennessee, Knoxville, TN, United States of America

* scarrign@utk.edu

## Abstract

The global pandemic of COVID-19 revealed the dynamic heterogeneity in how individuals respond to infection risks, government orders, and community-specific social norms. Here we demonstrate how both individual observation and social learning are likely to shape behavioral, and therefore epidemiological, dynamics over time. Efforts to delay and reduce infections can compromise their own success, especially when disease risk and social learning interact within sub-populations, as when people observe others who are (a) infected and/or (b) socially distancing to protect themselves from infection. Simulating socially-learning agents who observe effects of a contagious virus, our modelling results are consistent with with 2020 data on mask-wearing in the U.S. and also concur with general observations of cohort induced differences in reactions to public health recommendations. We show how shifting reliance on types of learning affect the course of an outbreak, and could therefore factor into policy-based interventions incorporating age-based cohort differences in response behavior.

## Introduction

Over the course of 2020, the numbers of COVID-19 cases rose, fell, and re-surged in many Western nations. This reflected the ineffectiveness in 2020 of many Western nations to 'flatten the curve' (Fig 3), amid substantial heterogeneity of infection levels and public and government responses. Groups repeatedly gathered unprotected in 2020, despite the public health recommendations against it [1].

More understanding is needed on the acceptability of different behavioral transmission interruption strategies in context with their effectiveness [2, 3]. With important lessons for future policy, the events of 2020 reflect a familiar challenge, in that the more successful a preventative strategy is (e.g. slowing an epidemic like COVID-19), the more public demand there is to relax those successful efforts before the threat has actually passed [4–6]. This recalls the 'Icarus paradox', or failure brought about by the same strategy that led to initial success [7]. Successful preventative public health measures can facilitate the illusion that they were

**Data Availability Statement:** All code used to generate the data is accessible here: http://github.com/simoncarrignon/slsir, compiled vignette is available here: https://simoncarrignon.github.io/

slsir/vignettes/paperPLOSONE.html and pre-run simulations results here: https://osf.io/fxjh4/.

**Funding:** This material is based upon work supported by the NSF under award #2028710. The funders had no role in study design, data collection and analysis, decision to publish, or preparation of the manuscript.

**Competing interests:** The authors have declared that no competing interests exist.

unnecessary in the first place. A virus such as COVID-19 is partly invisible as it is not only contagious without obvious symptoms [8] but inadequate testing of populations [9] meant that undocumented infections likely outnumbered documented infections by an order of magnitude [8–12]. These factors (along with mixed public messaging [13]) led to underestimated risk perceptions of the virus, which for many people were overshadowed by more transparent concerns, such as financial constraints [14, 15].

While the 'Icarus paradox' has been investigated in epidemiology and community medicine studies, the complexity lies in the multiple drivers of behavioural change, including information and social learning [16–19]. Ideally, decisions would be determined by their intrinsic payoffs within their socio-ecological environment [20]. In the real world, decisions are made by people who combine observational learning, which produces noisy information, and social learning, which diffuses that information to others [21, 22].

The transparency of learning is a crucial parameter [23]; the less transparent the payoffs are, the more noise and heterogeneity enters into the behavoral dynamics [24, 25]. While humans are motivated to avoid cues associated with pathogens, through emotions such as fear or disgust [26], COVID-19 and similar viruses display few cues in asymptomatic cases [27]. If the pathogen is less visible than the behaviors of peers, avoidance behaviors will be less effective at preventing the pathogen's spread. In this situation, most people would underestimate the threat based on personal observations of infected individuals. Hence, while general social distancing would be more protective than avoiding/isolating only those individuals with obvious symptoms, its benefits may be less transparent than its economic and psychological costs [15, 28, 29].

This suggests that drivers of social distancing include two dynamic factors: observable risks and observable behaviors. The risk of infection may not become apparent until the epidemic has become widespread. At the same time, social distancing that is more visible (e.g., mask-wearing or conspicuous lack of people in public spaces) may be adopted through social influence and conformity even before its benefits are widely observed [17, 30]. Subsequently, as the true benefits become more transparent to the public, individual cost-benefit decisions can support the behavior, in place of social conformity.

Here we frame the impact of the 'Icarus paradox' as dependent on information and timing, incorporating differential age-based cohorts impacts to reflect both increased influence from peers [31] and the well-publicized age-stratified risks from infection [32]. At what point do individuals observe enough infections around them to adopt preventative behavior as their own cost/benefit decision? Before this point, in the absence of government directives, social norms may be needed to encourage protective behaviors as people are not yet observing many infections. We expect that early in an outbreak, the link between protective behaviors and being uninfected will not yet be transparent. As infection prevalence increases, and the benefits become more obvious to individuals, social learning becomes less necessary to facilitate preventative behaviors.

We believe new insights are offered by modelling the interacting effects of noisy information and social learning on behavior. We take the approach of discrete behavioral choice with social influence [33–36], where we model decisions as based on a separable combination of two components: observational and social learning. Importantly, the underlying physical contact network through which a disease might spread [37–41], which can be mitigated by physical distancing, are different than the social learning pathways of information, behaviors, or beliefs [18, 41]. Social learning can take place via a variety of media and daily communicative activities, whether physically close or distant.

## Model

In order to model observational and social learning, we combine aspects of quantal response models [42, 43], which use a probabilistic (logit) choice function that allows errors in payoff estimation, with social learning of variable transparency [33, 44, 45]. Our model falls within the family of binary decision models with "positive externalities," meaning that the probability that one individual will choose a certain behavior will increase with the relative number of others choosing that same behavior [34, 46–48]. The difference with our model lies in the feedback between these externalities (social learning) with disease dynamics (observational learning).

As the dynamics are complex, we use an agent-based model (ABM) that explores two related dynamics: good protective behaviors reduce the transmission of the disease, whereas observing other infected individuals encourages (through observational learning) the adoption of protective behaviors. The aim is to tease apart the timing and impact of these different mechanisms for influencing individual behavioral choices over the course of an outbreak.

### Observational learning

The individual observational learning component is governed by the payoff difference between options, based on information people receive or observe to form beliefs about their health risks [17, 18, 41, 49, 50]. For example, the perceived utility of social distancing likely increases as more illnesses are observed. In a country like the U.S., where social distancing is complicated by numerous factors both political and socioeconomic [15, 51], we see nevertheless that the number of cases predicts a measure of social distancing such as mask-wearing. As shown in Fig 1a, the relationship between mask-wearing [52] and infections per square mile (from the second week of July 2020) is sigmoidal. This is consistent with typical formulations of discrete choice theory or quantal response theory, the probability that an agent chooses choice $i$ at time $t$ is proportional to $e^{\kappa U_i(t)} + \epsilon_i(t)$, where $\kappa$ is the transparency of choice and $U_i(t)$ is the intrinsic utility of the choice, and $\epsilon_i(t)$ is a noise term [33, 34, 42, 53].

Normalized across all choices, the sigmoid is shaped by two parameters. One parameter for this is the point at which one option has a higher utility than the other, $U_i > U_j$: we call this the 'inflection point', $v$. In our model, this inflection point represents the level of viral infections people perceive (assumed proportional to the number of infections in the age cohort) that signals the positive utility of social distancing in reducing infection risk. This average inflection point is a convenient modeling assumption; in reality there would be a distribution among the population as well as an 'identification problem' in detailing how a group's average behaviour influences its individuals [54, 55].

The second parameter for the observational learning component is transparency of choice, $\kappa$, which governs the steepness of the sigmoid curve: the higher the transparency of choice, the steeper the shape becomes, i.e. the smaller the variance in selecting the highest utility, $U$. In the real world, transparency might range from a clear, government-issued stay-home order (transparent), to seeing another person coughing, to a cacophony of conflicting media messages (non-transparent). Although these are quite different sources of information, they ultimately feed into an individual's estimate of their own risk. To a first approximation, this risk estimate from observational learning amounts to the infection rate around the person adjusted for the person's age.

We represent this as follows. Given $I_a$, the proportion of infected individuals in the cohort, $a$, the probability for an individual $i$ to switch from 'non-adherent' status, $NA$, to 'adherent'

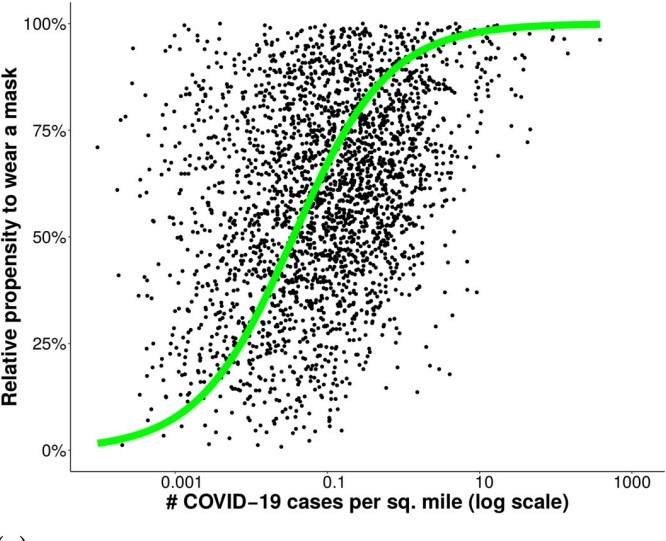

(a)

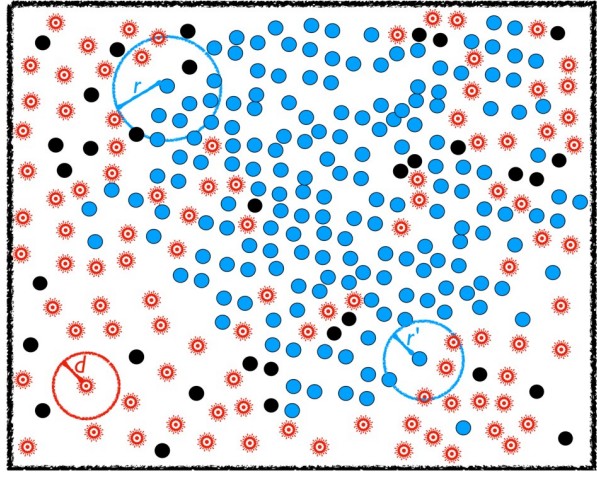

(b)

**Fig 1. Model and its rationale.** (a) Propensity of survey respondents in U.S. counties who 'Always' or 'Frequently' wear a mask in public during COVID-19 [52], versus confirmed infections per square mile. Data are from July 2020; each point represents a U.S. county. The green curve shows a sigmoid fit of $y = (1 + \exp[-1.6(x + 0.08)])^{-1}$. The 'propensity' on the y-axis accounts for the limited range in the county-aggregated data: about three-quarters of U.S. counties have survey results between 40% and 90% who 'always' wear masks, such that 'propensity to socially distance' measures how far each county plots between these endpoints. (b) Illustration of the agent-based model. Uninfected agents who are social distancing are colored black and agent who are not are colored blue. The circle in the lower left shows radius of infection, $d$.

status, $A$, is defined as:

$$P(a, NA \rightarrow A) \sim \frac{1}{1 + e^{-\kappa(I_a - v)}} \tag{1}$$

where $\kappa$ represents the transparency of choice—determining the steepness of the sigmoid curves—and $v$ defines the point at which the individual has a 50% probability to switch to adherent status, $A$. The influence of $v$ and $\kappa$, respectively, on the probability of adoption, $P$, is illustrated in Fig 2.

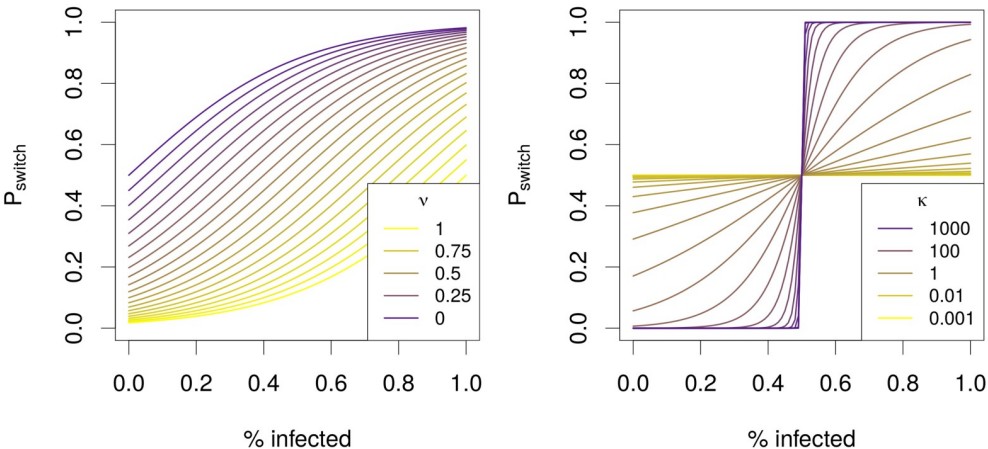

**Fig 2. Illustration of the impact of transparency of choice, $\kappa$, and point of inflection, $v$, on the decision curve.**

In reverse, the probability for an individual to switch back from adherent to non-adherent behavior, given the fraction of the age cohort who are not infected, $(1 - I_a)$, is:

$$P(a, A \rightarrow NA) \sim \frac{1}{1 + e^{-\kappa_r(1 - I_a - v_r)}} \qquad (2)$$

where $v_r$ is inflection point for reverting to $NA$ behavior (and can be different from $v$) and $\kappa_r$ the steepness of the curve use to calculate the probability of switch back to $NA$ decision (and can also be different from $\kappa$).

## Social learning

Social interactions play an important "role in mediating the spread of social contagions that impact health outcomes," [41]. Here we treat social distancing and related protective behaviors as 'simple' [41] contagions: copying what others do (e.g., wear a mask upon seeing someone else wear one). For our aggregated scale of modeling, we assume social influence operates 'as if' each person copies a currently healthy person, randomly encountered in the population, within a certain social or physical distance. The effect of the frequency of copied behaviors is stochastic, where probability of selecting option $i$ at a time $t$ is proportional to its current frequency or popularity, $p_{i,t}$ [33, 34, 54].

We also assume that social learning of simple behaviors (as opposed to complex skills learned over years) is biased by similarity, in that agents copy others within their cohort (a cohort could be age, political affiliation, etc). For convenience, this assumption conflates two observations, that: (a) cohorts tend already to share similar beliefs and preferences, derived from both ontogeny and broadly shared socio-economic landscape during early years of development [56–62], and (b) conformity tends to be age-dependent with cohort-biased social learning [63].

## Agent-Based Model (ABM)

The ABM (Fig 1b, Algorithms 1 and 2) consists of agents who are involved in two inter-related processes, the contagion of a disease in SIR fashion [64], and the spread of protective behaviors through a combination of individual and social learning. Agents move in a grid, with a normally-distributed range of speeds. The ABM is initialized with 500 agents and one infected agent. Within this space, an infected agent can infect an uninfected agent that is not socially

distancing if within 'distance' $d$ (Fig 1b). At the same time, agents of similar age are observing the levels of infection as well as protective behaviors, and adjusting their behaviors according to the observational and social learning rules described in Eq 1.

**Algorithm 1**: Main Algorithm

```
Data: N: number of of agents, t_step: number of time steps, P(b⃗): vector
      with the probability of infection for different behaviors, b d:
      the distance between two agents under which the disease can be
      transmitted, i_0: the number of initial infections, p: the proba-
      bility of observational learning, r: radius within which indi-
      vidual can learn socially κ: transparency (steepness of the
      sigmoid), ν: inflexion point of the sigmoid, κ_r: steepness of
      the reversion sigmoid, ν_r: inflexion point of the reversion
      sigmoid,
Result: A table with the SIR distribution per timestep
1 pop ← generatePopulation(N, x, y)
2 i ← pop[random(N)]
3 i.state ← "I" // randomly infect one individual
4 foreach t ∈ t_step do
5   foreach a ∈ pop do
6     a.move()
7     X ∼ U(0, 1)
8     if X < p then
        // Observational Learning
9       g_i ← getPropInfected(pop, a.age)
10      if i.behavior = "NA" then
11        p_switch ← sig(g_i, k, ν)
12        X ∼ U(0, 1)
13        if X < p_switch then a.behavior ← "A"
14      else
15        p_switch ← sig(1 - g_i, k_r, ν_r)
16        X ∼ U(0, 1)
17        if X < p_switch then a.behavior ← "NA"
18      end
19     else
        // Social Learning
20      m ← selectModel(pop, a.age, strategy, r) // select another
          individual to copy its behavior individual within radius r
21      a.behavior ← m.behavior
22     end
23     foreach i ∈ pop − a|dist(i, a) < d do
24       X ∼ U(0, 1)
25       if X < P[a.behavior] then a.state ← "I"
26     end
27     if a.state == "I" then
28       a.recovery ← a.recovery − 1
29       if a.recovery < 1 then a.state ← "R"
30   end
31 end
```

The algorithm of the ABM (Algorithms 1 and 2) relies on two main functions, one (*generatePopulation*) that generates the initial population of agents and the other (*selectModel*) that allows agents to copy the behavior of another agent, who must be both non-infected and within the same cohort, within radius *r* of the agent. Table 1 summarises its adjustable parameters.

Simulations were run with the parameters in Table 1, with the population initialised as described by the Algorithm 2. Some parameters were fixed ($N$, $d_i$, $i_0$). Initial behaviors were all

**Table 1. Adjustable parameters in the agent-based simulation, with 500 agents and initialized with one infected agent.**

| Parameter | Description | Value |
|---|---|---|
| $P_A$ | Probability a social-distancing agent (A) will infect within radius $d$ | 0.1 |
| $P_{NA}$ | Probability a non-social-distancing agent (NA) will infect within radius $d$ | 1 |
| $p$ | Probability to learn by individual observation | $U(0, 1)$ |
| $r$ | Radius of social learning | $U(1, 50)$ |
| $t_{step}$ | number of time steps | 1500 |
| $d$ | Maximum radius for disease transmission | 5 |
| $v, v_r$ | Inflection points of the learning functions | $U(0, 1)$ |
| $\kappa, \kappa_r$ | Transparency of choice for adopting & reverting | $10^{U(-1, 3)}$ |
| $R$ | Distribution of recovery times | $R \sim U(8, 14) \times 24$ |
| $S$ | Distribution of the speeds of the agents | $S = \mathcal{N}(1, 0.2)$ |

non-adherent to start each simulation. The speeds were sampled from a normal distribution $S \sim \mathcal{N}(1, 0.2)$ and recovery times were sampled from a uniform distribution $R \sim U(8, 14) \times 24$. The parameters were selected such that, overall, agents encounter a mean of 12 individuals per 24 time steps, such that each time step represents one hour. The recovery times in the SIR model were thus calculated to represent 8 to 14 days as a reasonable approximation for COVID-19 [65].

To explore the impact of learning on the disease outcomes over the course of an outbreak, we ran three sets of models. We first ran two baseline conditions in which individual behavior is held constant as either non-adherent (worst case scenario) or adherent (best case scenario) over the course of the simulation. We subsequently ran the models with the parameters described in Table 1 after initializing the population following Algorithm 2. In all, we ran 826,892 simulations and recorded the number of infected agents at each time step in each simulation.

**Algorithm 2**: Generate a population of agents

```
Data: N: number of agents, R: a uniform distribution of recovery
      times, B: a distribution of behaviors, S: a normal distribution
      of speeds, x,y: spatial limits,
Result: a table with n agents
1 pop = foreach a ∈ n do
2   a.age ~ {.24, .09, .12, .26, .13, .16} × n // source: https://www.
    kff.org/other/state-indicator/distribution-by-age/
3   a.behavior ← "NA" // by default start with everyone is Non-
    adherent
4   a.position ← (rand(x), rand(y)) // random position in the grid
5   a.state ← "S"
6   a.recovery ~ U(8, 14) // agents have different recovery times,
    sampled between 8 and 14 days
7   pop.add(a)
8 end
9 return(pop)
```

To summarize our simulations in a way that captures the 'flattening the curve', we focused on two metrics: (a) the maximum number of infected people $I_{max}$, and (b) the time to reach this maximum $\tau$. As in Fig 3, the 'flat' curve has lower $I_{max}$ and larger $\tau$. Across all simulations, we recorded the largest and smallest maximum total infections among all the runs, as max$(I_{max})$ and min$(I_{max})$, respectively, as well as the longest and shortest times to reach the max, as max$(\tau)$ and min$(\tau)$, respectively. Since these dimensions will tend to be inversely correlated,

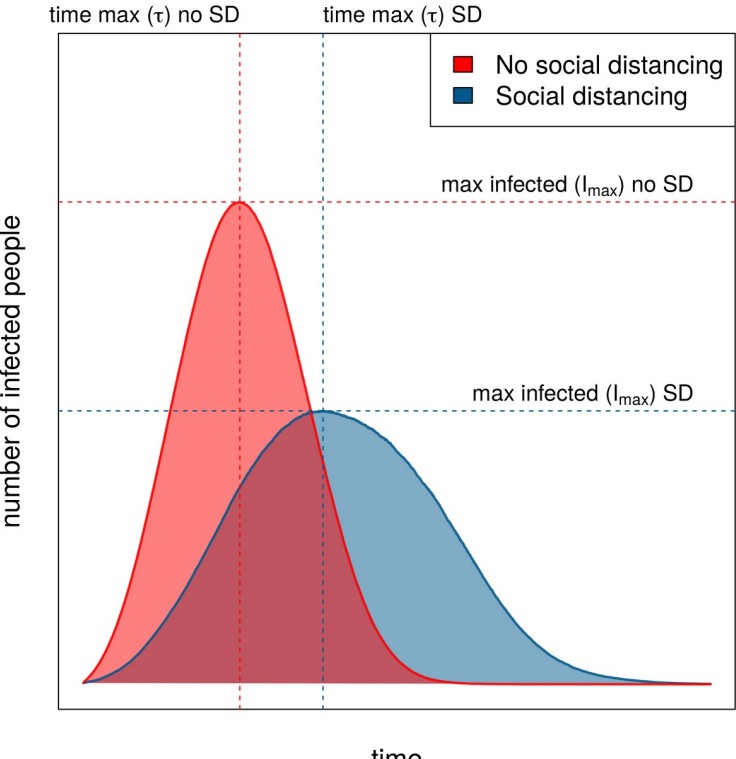

**Fig 3. Illustration of the impact of social distancing on the spread of the virus.**

we defined a metric, $\delta(s)$, for a set of simulations $s$ summarizing both:

$$\delta(s) = \frac{1}{2}\left(1 - \frac{\tau - \min(\tau)}{\Delta\tau} + \frac{I_{max} - \min(I_{max})}{\Delta I_{max}}\right) \tag{3}$$

where $\Delta I_{max} = [\max(I_{max}) - \min(I_{max})]$ and $\Delta\tau = [\max(\tau) - \min(\tau)]$, from all simulations.

Whereas our results below involve the simulation just described, we also explored three alternative scenarios, described in the S1 Text. In the first scenario individuals could copy any other individual via social learning, rather only those seen to be healthy. In the second scenario, part or all of the initial population already adhere to social distancing. The third scenario allows learning only after a certain number of time steps. The results yielded by those three alternative scenarios are given in the S1 Text but do not differ from the one presented in the next section.

## Results and discussion

Our simulations produced meaningfully different outcomes that varied in the success of 'flattening the curve' (Fig 4, left). This reflects the expected patterns of outbreak curves under the impact of social distancing. Our simulations reveal multiple possible outcomes of behavioral mitigation, from 'flat' infection curves to full, unchecked outbreaks (Fig 4, right), some of which are direct illustration of the Icarus paradox.

Which learning parameters yielded the 'flattest' curve? The simulations reveal a mix of parameters underlying the best preventative outcomes. Among the simulated outbreaks in which $\delta$ were minimized—what we call the 'best' outcome—the transparency and inflection

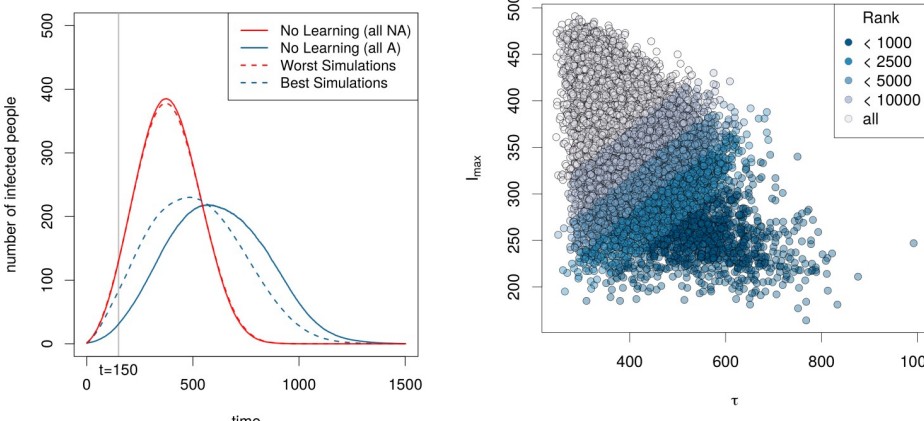

**Fig 4. Left**: Comparing the mean trajectories of SIR model where agents do not learn and instead continue in their initial behavior, with trajectories generated by simulations. Red curve: all agents are non-adherent to social distancing and stick to this behavior. Blue curve: all agents adhere to social distancing. Dashed red line: mean trajectory of the 1, 000 worst simulations. Blue dashed line: mean trajectory of simulations run with the parameters selected in the 1, 000 best simulations with regard to $\delta$. Vertical gray line: status after 150 time steps. **Right**: Distribution of 826, 892 simulations given two different metrics: the maximum number of infected people (y-axis) and the time to reach this maximum (x-axis). The colors represent different class of the simulations ranked given $\delta$, the metric defined in Eq 3.

points ($v_r$ and $\kappa_r$) for reversion to non-protective behavior needed to be more constrained than the respective parameters ($v$ and $\kappa$) for choosing protective behavior. Fig 5 shows that the range of optimal parameters are more constrained by the end of the simulation (bottom row), which represents about two months, than they were after just 150 time steps (top row), representing approximately one week (see gray line in Fig 4). In the S1 Fig shows these joint posterior distributions in more detail.

Focusing on the end of the simulation (Fig 5, bottom row), we see different ranges of parameters yielding the 'best' (blue) and 'worst' (red) outcomes. Fig 5 shows with the joint posterior distributions of $v$ versus $\kappa$ (adopting protective behaviors) and of $v_r$ versus $\kappa_r$ (reversion back) on the left and right, respectively. Note particularly that the parameter regions flip-flop between adopting and reverting back. In other words, for the 'best' outcome, it is better to have low $\kappa$ for adopting protective behaviors but then high $\kappa$ for reverting back to non-protective behavior (see the cross-sectional distributions along the axes of each bi-variate plot in Fig 5, lower row). At the same time, for the 'best' outcome $v$ can be virtually any value for switching to protective behaviors, but needs to be quite high for reverting back. This exemplifies the 'Icarus' paradox: when preventive actions triggered by environmental cues bring about improvements that subsequently trigger premature relaxation of those preventative behaviors.

As a qualitative interpretation of the asymmetric effects of $v$, the desired 'flat' curve is achieved through *low* transparency of choice in adopting the protective behavior, but high transparency about whether to revert back. Qualitatively the same is true of the inflection point, $\kappa$, which needs to be low in adopting the behavior but high in reverting back, in order to yield a 'flat' infection curve. These results suggest the 'best' outcomes involved a 'ratchet' effect: a low, fuzzy barrier to adopt the behavior but a high, sharp (transparent) barrier to revert back.

Our results imply that combined observational and social learning can drive successful mitigation strategies (Fig 4), even with differential, age-based disease risk. In an 'Icarus' scenario, where an initially successful strategy can lead to its own failure, the timing of these forms of learning are crucial. Those simulations yielding a flat curve generally required either low transparency of choice, low inflection point for adopting the behavior, and/or high inflection point

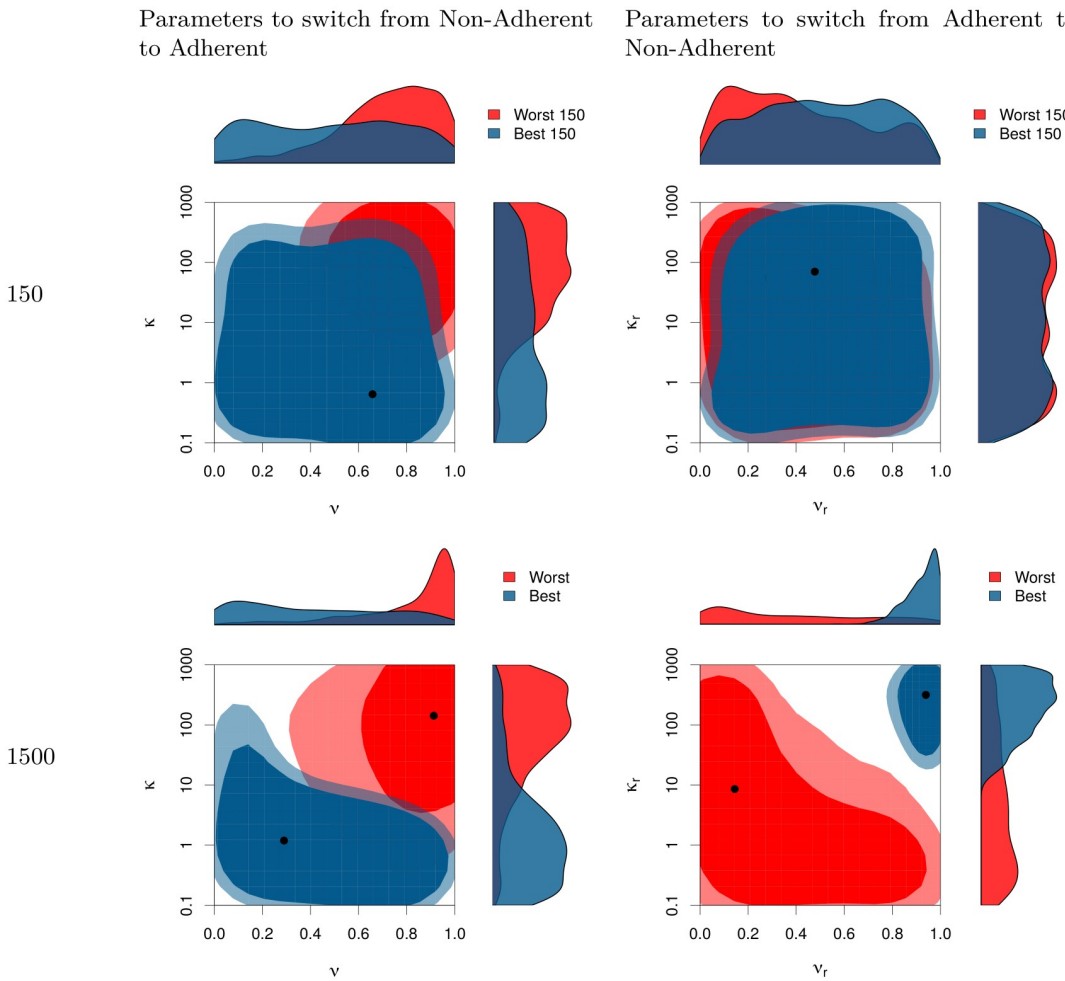

**Fig 5. Joint posterior distributions for the parameters used to switch from Non-Adherent to Adherent (left column) and for the parameters used to switch back from Adherent to Non-Adherent (right column).** The two-dimensional areas represent the 70% and 90% HDR (High Density Region) *i.e.*, the smallest areas within which respectively 70% and 90% of the parameters combination fall (for the mathematical definition of HDR and how they can be represented see [66]. Lighter colors represent the 90% HDR whereas darker represent the 70% HDR. The top row represents the value for those parameters that minimize (in blue) or maximize (in red) the number of infected people at time step 150. The bottom row shows the parameters that minimize (in blue) or maximize (in red) $\delta$ (as defined in Eq 3) at the end of the simulation (1500 time steps). Marginal posteriors for each parameter are drawn in the margin.

for reversion back to non-protective behavior—in other words, a reluctance to give up the protective behavior is more important than a low threshold to acquire it (Fig 5). While it is not surprising that strong maintenance of risk-averse behaviors would protect a population, the low, fuzzy barrier to adopt these behaviours is notable and shows parallels to theory on the diffusion of innovations [48, 67, 68]. The challenge is that the risk needs to sufficiently observable to the populations that need to undertake meaningful chance. Hence protective behaviors may be not be adopted until infection rates are very high, especially if most symptomatically infected individuals are not be publicly visible (e.g., remaining at home or in hospital, or else segregated by age).

For this reason, low transparency of choice—such as poor or conflicting information —can 'jump-start' adoption of protective behaviors by stretching out the inflection into a range, such

that some individuals 'mis-estimate' infection risks as enough to trigger their decision (see Fig 5). However, this result relies on the breadth of the distribution of responses to low transparency, rather than an alternative case in which either leadership or social norms cause low transparency to lead to greater average hesitancy to take any action [69].

Social learning dynamics are crucial, as "copying recent success" is often highly adaptive [70]. If symptoms/prevalence of a disease are transparent, copying healthy individuals ('success') should increase protective behaviors. Because COVID-19 can be asymptomatic, however, infected individuals may seem healthy, such that non-protective behaviors can be copied even through a "copy success" strategy. This lack of transparency may critically compromise learning-only strategies for successful disease risk containment. By being aware of this effect, policies could be better crafted to forestall eventual rejection and thereby remain more effective in the longer term.

These effects may be heightened among young people, for whom COVID-19 infection is more frequently asymptomatic [27] and who are also socially influenced by peers in their cohort [63]. Hence, while lower fatality risk from COVID-19 may help explain why some groups of young people rejected social distancing early in the U.S. outbreak [1], another important dynamic has likely been social conformity and lack of transparency about infections among their peers.

One limitation in our simulations is that we employed a broad definition of observational learning, as agents estimate infection rates with some degree of precision (transparency). Observational learning is heterogeneous; at any given moment, each person observes different information and a different segment of the population. The real world encompasses myriad information sources, individual experiences, and biases. Also, information from media will have a different effect on decision-making than having friends and family become ill [71]. Individuals who follow stay-home protocols have different interactions than those who interact in public spaces. Taken together, these processes affect the transparency of how observably protective behaviors relate to disease risks.

These considerations are particularly relevant for mitigating pandemic spread where strong governmental control is not possible. When individual choice is the driving factor in protective behaviors, cohort effects become important since different demographic segments of societies are likely to be more initially risk-averse than others. In terms of age cohorts, for example, older individuals may be early adopters of adherent behaviors because news reports of mortality rates in older populations create a psychological burden of fear. By contrast, younger and/or more economically limited individuals may delay switching to behavioral adherence, doing so only when they feel their circumstances allow it. In such cases, the socioeconomic and demographic representation of a population may be the critical drivers in determining whether individual behaviors, guided by learning, can be relied upon to effectively achieve outbreak mitigation by adherence to social distancing.

In our models, we have also assumed that choices are rational and focused only on epidemiological risk. Human decision-making is never fully rational, especially during period of stress [72]. Further, we have here explored only one potential route of social learning, in which individuals simply copy the behaviors of perceived healthy individuals. More nuanced approaches may emphasize preferentially copying people who share your beliefs, alignment with deeply-held beliefs or values, and other social learning strategies [41, 73–75]. Similarly, we have construed frequency of contact within a spatial radius to be the only medium for social derivation of learning. In reality, of course, individuals have many means of arriving at their sense of what other people are doing, including unique personal experiences and personal choice of media, however complicated or accurate [41, 76–78]. This raises many interesting questions, such as the different effect of centralized versus diverse media, for example. Given our

parsimonious starting point, future expansion and testing of our models can address these potentially important features of social groups.

We also focused on public health measures that could be quickly adopted, in the initial case where sweeping lockdowns are not politically feasible. Our model examines the initial transient coupled dynamics directly following the introduction/emergence of a novel pathogen threat. The model proposed here focuses exclusively on the first, major wave of infections. While our contribution has been to couple the social and individual responses, we note that even in an asocial awareness model, a pandemic peak can become a prolonged plateau rather than a sharp peak, if individual transmission decreases in response to awareness of the disease at a population level [16, 79, 80].

The past two years have witnessed multiple surges of COVID-19 in many countries, including the U.S., for a diverse range of socio-political reasons [81]. Our base model presented here — focused on a single coherent, population-level response— would not, on its own, explain the observed secondary waves of COVID-19. In 2020–2021, different municipalities quickly diverged in their response to COVID-19 risks; lockdowns lasted different durations of time [79], political discourse became divisive [81], and later new strains such as Delta altered the disease epidemiology. In a mathematical model, one way to induce a second wave is to introduce a step function, representative of a centralized decision to remove lockdowns (i.e. increase spread) when public support for closure drops below a certain threshold. [82] introduce such a model, focused on centralized decisions such as school closures, that also includes social learning dynamics underlying these centralized "on-off" decisions. Since our model is based on individual, rather than centralized, decisions, this step-change instigating a second wave would not occur in our base model without more complex modifications, although this would be a valuable topic for future research. In particular, we anticipate future work where studies build on our work to investigate the various reactions that could occur as outbreaks progress. Future exploration of the model could integrate these aspects, by adding asymptomatic individuals, the apparition of new variants, more complex population structure and the possibility for recovered individuals to be infected again after a certain time. We could then explore which social learning features can mitigate subsequent waves.

Understanding how the timing and dynamics of different types of learning affect individual behavior over the course of an outbreak, as disease prevalence changes the transparency of benefits of protective behaviors, while those behaviors become more visible as they proliferate. Social learning can help boost protective behaviors, but not until the number of infections has brought about those behaviors initially through observation-driven decision making. For diseases such as COVID-19, in which age-based differences in observability and risk complicate age cohort-based learning, it may be particularly important to understand the dynamics of observability over time.

These insights translate directly into concrete recommendations for how to communicate protective health policies. Our results indicate that social influencers can be recruited from different demographic cohorts to help model protective behaviors in populations. The effect may be most poignant when certain cohorts have become jaded by or resistant to public health efforts—the point our simulations indicate can drastically shift the course of an outbreak. Further, a shift away from messages of communal support (e.g. "We're all in this together") to messages that more closely reflect cheer-leading (e.g. "It's working, people!") might meaningfully alter adoption and/or maintenance of protective behaviors. Making protective behaviors publicly visible among different cohorts should help to sustain those increases by social influence, and help attenuate the ensuing complacency and reversion to non-protective behavior. Policy makers who incorporate this perspective of social learning can formulate more effective

policies to anticipate the potential self-defeating effects of initially successful public protective behaviors in a disease outbreak.

## Supporting information

**S1 Fig. All posterior distributions.** All posterior distributions for the 6 parameters used in our simulations. The white area represents the distribution of the parameters of all simulations (the priors), the blue area represents the distribution for the top 1000 simulations rank using $\delta$ while the shaded area represent the parameters of the 1000 simulations with the lowest number of infected people after 150 timesteps.Left column represents the parameters value to switch from Non-Adherent to Adherent, middle column from Adherent to Non-Adherent, and right column the paramaters that defines the probability and the radius of social learning.
(TIFF)

**S2 Fig. Joint posteriors for different levels of selection.** Joint posteriors of representative pair of parameters of the model for different level of what we consider as the "best" simulations. Each distribution represent the parameter distribution of simulations for which the metric $\delta$ is ranked below different level, from top to bottom: $10, 000; 5, 000; 2, 500; 1, 000$. The 2d areas represent the High Density Regions, ie the smallest regions within which falls a certain percentage of the distribution of the parameters our selected simulation. The lighter areas represent the area within which all the simulations are distributed and darker areas represent regions for smaller HDR.
(TIFF)

**S1 File. Data and code.** All data generated or analysed during this study are included in this published article (and its Supporting Information files). The code used to implement, run and analyse models is in R3.5 [83] and available at: https://github.com/simoncarrignon/slsir.
(TXT)

**S1 Text. Different model of social learning and initial distribution.** This text describes and illustrates results for 4 variants of the original model.
(PDF)

## Acknowledgments

This material is based upon work supported by the NSF under award #2028710.

## Author Contributions

**Conceptualization:** Simon Carrignon, R. Alexander Bentley, Matthew Silk, Nina H. Fefferman.

**Formal analysis:** Simon Carrignon, R. Alexander Bentley.

**Funding acquisition:** Nina H. Fefferman.

**Software:** Simon Carrignon.

**Visualization:** Simon Carrignon, Matthew Silk.

**Writing – original draft:** Simon Carrignon, R. Alexander Bentley, Matthew Silk, Nina H. Fefferman.

**Writing – review & editing:** Simon Carrignon, R. Alexander Bentley, Matthew Silk, Nina H. Fefferman.

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
