## [Decision Letter · Decision Letter 0]

14 Sep 2021

PONE-D-21-16649How Social Learning Shapes the Efficacy of Preventative Health Behaviors in an Outbreak

PLOS ONE

Dear Dr. Carrignon,

Thank you for submitting your manuscript to PLOS ONE. After careful consideration, we feel that it has merit but does not fully meet PLOS ONE’s publication criteria as it currently stands. Therefore, we invite you to submit a revised version of the manuscript that addresses the points raised during the review process.

We look forward to receiving your revised manuscript.

Kind regards,

Ceyhun Eksin

Academic Editor

PLOS ONE

“This material is based upon work supported by the NSF under 337 award #2028710”

Additional Editor Comments (if provided):

I received two reviews who both commend the effort to include social learning theories in modeling public behavior during an epidemic. They also highlight a few major issues that needs to be addressed in the revised manuscript.

- Reviewer 1 identifies that the supporting documents and code are missing, and hard to follow/replicate.

- Reviewer 1 points that the protective behavior adopted by sick individuals can reduce the chance of spread and suggests including such effects can further increase the chance of observing the Icarus Paradox.

- Reviewer 2 provides a list of recent publications that incorporate behavioral changes in epidemic models. These behavioral models are either mechanistic or are based on theories in behavioral economics.

- Both reviewers suggest improving the relevance of the model to COVID-19. Specifically, Reviewer 2 questions the relevance of results to COVID-19 where multiple peaks are observed.

I concur with the reviewers' assessments. I would also urge the authors to discuss their model in light of some of the existing and recent epidemic models that incorporate social behavior.

Reviewers' comments:

Reviewer's Responses to Questions

**Comments to the Author**

1. Is the manuscript technically sound, and do the data support the conclusions?

Reviewer #1: Yes

Reviewer #2: Partly

2. Has the statistical analysis been performed appropriately and rigorously? 

Reviewer #1: Yes

Reviewer #2: Yes

3. Have the authors made all data underlying the findings in their manuscript fully available?

Reviewer #1: No

Reviewer #2: Yes

4. Is the manuscript presented in an intelligible fashion and written in standard English?

Reviewer #1: Yes

Reviewer #2: Yes

5. Review Comments to the Author

Reviewer #1: The authors use agent based models to gain insight into what drives the Icarus Paradox (where public health interventions which initially work well become less effective over time), modeling how the addition of social learning impacts the adoption of preventative behaviors. The authors augment an SIR model with two types of social learning: Observational Learning (copying from members of their age cohort) and Social Learning (copying from other agents physically close). Importantly in this model agents are allowed to transition from good behavior to bad behavior, which is very relevant to the current course of the COVID-19 pandemic. The authors propose a metric for measuring curve flatness and demonstrate how key model parameters influence curve flatness. The model the authors propose is a reasonable epidemic model. Given the calibration and framing around the COVID-19 pandemic, my suggestions are geared towards improving generalizability to the COVID-19 pandemic. Below are my suggestions for revision:

Major Comments:

1. The replication package seems to be missing files required for all of the included scripts to run. I could not find abmEpi.R and covid.masks_small.csv in the linked github repository and I could not find S1 File in the list of supplements I got. I could only get the Intro vignette to run and was not able to get the other vignettes used for replicating the graphs to run. I was able to get slsirSimu() and generatePopulation() to work as expected. The visualizations provided by the visu flag were well designed and informative.

2. Having the behavioral state impact how effectively infected individuals spread the pathogen could help the model better generalize to the COVID-19 pandemic. While there are some behavioral interventions that prevent infection in the person doing them (self isolation), many of these behaviors also make it less likely for the infected to spread the disease (mask wearing). My hunch is that this would increase the likelihood of observing the Icarus Paradox.

3. I would be interested to see how sensitive the results are to the parameter Pa. If Pa were higher, then more infections would “breakthrough” the social distancing and there would be more infections even with high levels of social distancing adoption. This could impact how the Icarus effect manifests. It could strengthen the results to show how social learning impacts behavioral choice, when the behaviors have different levels of efficacy in reducing spread.

Minor Comments:

1. The slsirSimu() and generatePopulation() are private which can make the less user friendly for running your own simulations with different parameters

2. There are some typos in the documentation for the functions and the package metadata is still the default package metadata.

3. It would be helpful to clarify which scripts / markdown documents are essential for replicating the results in the paper (especially the scripts in the exec folder).

Reviewer #2: In this paper, authors examine the coupled evolution of the SIR epidemic and social learning among individuals in an agent-based framework. The authors propose that the strategies that led to initial success in containing COVID might have led to emergence of larger subsequent waves of infection; a phenomenon known as the Icarus Paradox. In the proposed model, agents choose to adopt protective behavior (such as wearing masks) in a logit choice framework based on two main factors:

(i) observational learning: this capture adoption and termination of protective behavior as a function of perceived infection level

(ii) social learning: copy the behavior of an individual in a similar cohort.

The authors carry out large scale agent based simulations to show the effect of social learning on epidemic evolution for different inflection points; captured via peak infection level and time to peak infection. The plots signify various degrees of curve flattening due to the positive externality induced by social learning.

However, my major concern is the lack of plateaus and secondary waves in the simulations as is observed in real epidemics. In fact, many recent papers have observed that it is precisely the behavioral factors and joint evolution of human response and epidemics that lead to emergence of second waves and plateaus. Some recent papers are mentioned below.

[1] Weitz, Joshua S., et al. "Awareness-driven behavior changes can shift the shape of epidemics away from peaks and toward plateaus, shoulders, and oscillations." Proceedings of the National Academy of Sciences 117.51 (2020): 32764-32771.

[2] Toxvaerd, Flavio MO. "Equilibrium social distancing." University of Cambridge, Working Paper (2020).

[3] McAdams, David. "The Blossoming of Economic Epidemiology." Annual Review of Economics 13 (2021).

[4] Pedro, Sansao A., et al. "Conditions for a second wave of COVID-19 due to interactions between disease dynamics and social processes." Frontiers in Physics 8 (2020): 428.

It will be important to explore why the proposed model does not capture these important phenomena. If it can be shown that social learning or lack of transparency that led to early containment of first wave, but immediate reverting to non-adherence led to emergence of second waves, then the proposed model would be better justified and the paper would be stronger.

6. PLOS authors have the option to publish the peer review history of their article (what does this mean?). If published, this will include your full peer review and any attached files.

Reviewer #1: No

Reviewer #2: No

---

## [Author Response · Author response to Decision Letter 0]

3 Nov 2021

Manuscript Number: PONE-D-21-16649

“How social learning shapes the efficacy of preventative health behaviors in an outbreak”

Reviewer’s comments are in red, our responses in black, and changes to manuscript in blue. 

Reviewer #1: 

1. The replication package seems to be missing files required for all of the included scripts to run. I could not find abmEpi.R and covid.masks_small.csv in the linked github repository and I could not find S1 File in the list of supplements I got. I could only get the Intro vignette to run and was not able to get the other vignettes used for replicating the graphs to run. I was able to get slsirSimu() and generatePopulation() to work as expected. The visualizations provided by the visu flag were well designed and informative.

==> A new vignette complete has been added to the repository and a compiled version can be accessed through https://simoncarrignon.github.io/slsir/vignettes/paperPLOSONE.html. This should allow anyone to reproduce all figures and analysis of the paper.

2. Having the behavioral state impact how effectively infected individuals spread the pathogen could help the model better generalize to the COVID-19 pandemic. While there are some behavioral interventions that prevent infection in the person doing them (self isolation), many of these behaviors also make it less likely for the infected to spread the disease (mask wearing). My hunch is that this would increase the likelihood of observing the Icarus Paradox.

==> Agreed. These kinds of behavioral variations are part of ongoing work, where we try to build on the simple foundation of the model in this paper.

 I would be interested to see how sensitive the results are to the parameter Pa. If Pa were higher, then more infections would “breakthrough” the social distancing and there would be more infections even with high levels of social distancing adoption. This could impact how the Icarus effect manifests. It could strengthen the results to show how social learning impacts behavioral choice, when the behaviors have different levels of efficacy in reducing spread.

==> We address this with tests (see below), but note that it effectively nullifies our research question on the effect of heterogeneous social distancing on the spread of a pandemic. Increasing Pa reduces the efficiency of social distancing, and hence the phenomenon we intended to measure disappears. To illustrate it we have simulated two additional model variants, labelled H and G (supplementary material), where Pa is respectively 0.2 and 0.6. The posterior distributions of the switching parameters illustrates well how the adoption mechanism become less important as the curves become more similar to the priors (Figure 8 and 9 in S1_text.pdf), whereas the overall number of infected individual increases (Fig1 in S1 text, curves G and H), tending toward cases where flattening the curve through social learning becomes impossible.

Minor Comments: 1. The slsirSimu() and generatePopulation() are private which can make the less user friendly for running your own simulations with different parameters 2. There are some typos in the documentation for the functions and the package metadata is still the default package metadata. 3. It would be helpful to clarify which scripts / markdown documents are essential for replicating the results in the paper (especially the scripts in the exec folder).

==> Regarding all three minor points: The package has been updated. The main functions are now exported and public. The documentation has been clarified. Scripts in the exec/ folder have been translated as vignettes with clearer explanations. 

Reviewer #2: 

However, my major concern is the lack of plateaus and secondary waves in the simulations as is observed in real epidemics. In fact, many recent papers have observed that it is precisely the behavioral factors and joint evolution of human response and epidemics that lead to emergence of second waves and plateaus. …. [1] Weitz, Joshua S., et al. "Awareness-driven behavior changes can shift the shape of epidemics away from peaks and toward plateaus, shoulders, and oscillations." Proceedings of the National Academy of Sciences 117.51 (2020): 32764-32771. [2] Toxvaerd, Flavio MO. "Equilibrium social distancing." University of Cambridge, Working Paper (2020). [3] McAdams, David. "The Blossoming of Economic Epidemiology." Annual Review of Economics 13 (2021). [4] Pedro, Sansao A., et al. "Conditions for a second wave of COVID-19 due to interactions between disease dynamics and social processes." Frontiers in Physics 8 (2020): 428. It will be important to explore why the proposed model does not capture these important phenomena. If it can be shown that social learning or lack of transparency that led to early containment of first wave, but immediate reverting to non-adherence led to emergence of second waves, then the proposed model would be better justified and the paper would be stronger.

==> These are excellent studies, all of which we now cite in context (see below). Clearly, progress is being made on multiple fronts, and our study constitutes one among several different approaches. Regarding a second wave, this can be modelled via a step-function representing a system-wide change in social distancing (e.g. lockdowns or school closures), whereas our individual-based model does not have this mechanism. Hence this phenomenon would require additional parameters and a full re-implementation of the model, which is outside the scope of current work (although we would be interested to apply in future). In response to these important points, we have added the following paragraphs to our Discussion:

" Here we have focused on public health measures that could be quickly adopted, in the initial case where sweeping lockdowns are not politically feasible. Our model examines the initial transient coupled dynamics directly following the introduction/emergence of a novel pathogen threat. The model proposed here focuses exclusively on the first, major wave of infections. While our contribution has been to couple the social and individual responses, we note that even in an asocial awareness model, a pandemic peak can become a prolonged plateau rather than a sharp peak, if individual transmission decreases in response to awareness of the disease at a population level (Funk et al. 2010; Weitz et al. 2020; Toxvaerd 2020). 

The past two years have witnessed multiple surges of COVID-19 in many countries, including the U.S., for a diverse range of socio-political reasons (McAdams 2021). Our base model presented here— focused on a single coherent, population-level response— would not, on its own, explain the observed secondary waves of COVID-19. In 2020-2021, different municipalities quickly diverged in their response to COVID-19 risks; lockdowns lasted different durations of time (Weitz et al. 2020), political discourse became divisive (McAdams 2021), and later new strains such as Delta altered the disease epidemiology. In a mathematical model, one way to induce a second wave is to introduce a step function, representative of a centralized decision to remove lockdowns (i.e. increase spread) when public support for closure drops below a certain threshold. Pedro et al. (2020) introduce such a model, focused on centralized decisions such as school closures, that also includes social learning dynamics underlying these centralized “on-off” decisions. Since our model is based on individual, rather than centralized, decisions, this step-change instigating a second wave would not occur in our base model without more complex modifications, although this would be a valuable topic for future research. In particular, we anticipate future work where studies build on our work to investigate the various reactions that could occur as outbreaks progress. Future exploration of the model could integrate these aspects, by adding asymptomatic individuals, the apparition of new variants, more complex population structure and the possibility for recovered individuals to be infected again after a certain time. We could then explore which social learning features can mitigate subsequent waves.

"

---

## [Decision Letter · Decision Letter 1]

27 Dec 2021

How Social Learning Shapes the Efficacy of Preventative Health Behaviors in an Outbreak

PONE-D-21-16649R1

Dear Dr. Carrignon,

We’re pleased to inform you that your manuscript has been judged scientifically suitable for publication and will be formally accepted for publication once it meets all outstanding technical requirements.

Kind regards,

Ceyhun Eksin

Academic Editor

PLOS ONE

Additional Editor Comments (optional):

Thank you for successfully addressing the comments of both of the reviewers.

Reviewers' comments:

Reviewer's Responses to Questions

**Comments to the Author**

1. If the authors have adequately addressed your comments raised in a previous round of review and you feel that this manuscript is now acceptable for publication, you may indicate that here to bypass the “Comments to the Author” section, enter your conflict of interest statement in the “Confidential to Editor” section, and submit your "Accept" recommendation.

Reviewer #1: All comments have been addressed

Reviewer #2: All comments have been addressed

2. Is the manuscript technically sound, and do the data support the conclusions?

Reviewer #1: Yes

Reviewer #2: Yes

3. Has the statistical analysis been performed appropriately and rigorously? 

Reviewer #1: Yes

Reviewer #2: Yes

4. Have the authors made all data underlying the findings in their manuscript fully available?

Reviewer #1: Yes

Reviewer #2: Yes

5. Is the manuscript presented in an intelligible fashion and written in standard English?

Reviewer #1: Yes

Reviewer #2: Yes

6. Review Comments to the Author

Reviewer #1: The authors have addressed all of my concerns. While testing I did a fresh install of the package, github install did not work since there is a missing comma in the description file between the two author names. When I fixed that, doing a local install and the rest of the changes you made to the package worked fine.

Reviewer #2: Thank you for adding relevant details and discussions in response to the reviewer comments. I have no further comments.

7. PLOS authors have the option to publish the peer review history of their article (what does this mean?). If published, this will include your full peer review and any attached files.

Reviewer #1: No

Reviewer #2: No

---

## [Editor Report · Acceptance letter]

3 Jan 2022

PONE-D-21-16649R1 

How Social Learning Shapes the Efficacy of Preventative Health Behaviors in an Outbreak 

Dear Dr. Carrignon:

I'm pleased to inform you that your manuscript has been deemed suitable for publication in PLOS ONE. Congratulations! Your manuscript is now with our production department. 

Kind regards, 

on behalf of

Dr. Ceyhun Eksin 

Academic Editor

PLOS ONE